# A Large-Scale Outbreak of Trichinellosis from Infected Wild Boar Meat in Croatia and the Role of Real-Time PCR Assays in Confirming the Source of the Disease

**DOI:** 10.3390/microorganisms11122995

**Published:** 2023-12-16

**Authors:** Davor Balić, Tomislav Dijanić, Marija Agičić, Josip Barić, Maria Kaltenbrunner, Hrvoje Krajina, Rupert Hochegger, Mario Škrivanko, Karlo Kožul

**Affiliations:** 1Croatian Veterinary Institute, Department Vinkovci—National Reference Laboratory for Parasites (Genus Trichinella), 32100 Vinkovci, Croatia; marija.agicic@gmail.com (M.A.); cajperzuko@gmail.com (H.K.); skrivanko@veinst.hr (M.Š.); 2Public Health Institute of Osijek-Baranja County, 31000 Osijek, Croatia; dijanict@gmail.com (T.D.); dr.kkozul@gmail.com (K.K.); 3State Inspectorate of the Republic of Croatia, Osijek Regional Office—Vinkovci Branch Office, 32100 Vinkovci, Croatia; josip.baric@dirh.hr; 4AGES—Austrian Agency for Health and Food Safety, Institute for Food Safety Vienna, 1210 Vienna, Austria; maria.k@edumail.at (M.K.); rupert.hochegger@ages.at (R.H.)

**Keywords:** trichinellosis, wild boar, real-time PCR, Croatia, meat products, *T. spiralis*

## Abstract

Background: Trichinellosis in Croatia posed a significant health concern during the 1990s, followed by a notable improvement in the epidemiological situation. However, in 2017, there was a resurgence, with 37 recorded cases in 3 outbreaks and 3 sporadic cases. The source of this epidemic was homemade meat products derived from wild boar meat, leading to 26 infections. Methods: At the beginning of the outbreak and during the treatment of the patients, the medical and epidemiological records prepared throughout the investigation and over the course of patient treatment were reviewed. The recovery of the first-stage (L1) larvae from suspect meat products was achieved by artificial digestion. The molecular identification of the isolated larvae was performed by multiplex PCR. The molecular identification of the meat used to prepare the meat products was performed by real-time PCR assays. Results: The epidemic started in early 2017. In total, 71 exposed persons were documented: 26 with clinical symptoms and 3 hospitalised in two cities in different counties. The L1 burden in three different meat products was from 5.25 to 7.08 larvae per gram (LPG), and *T. spiralis* was determined as the aetiological agent of the outbreak. The molecular and biological identification confirmed that implicated meat products were made solely from wild boar meat. Conclusions: Although trichinellosis is no longer a frequent occurrence in Croatia, several cases are still registered nearly every year. Wild boar meat poses an important risk factor for human health if compulsory testing is not conducted before consumption, especially if the meat products are consumed without proper thermal processing.

## 1. Introduction

Trichinellosis can be a serious or even deadly disease caused by a parasite from the genus *Trichinella*.

In over 60% of reported epidemics, insufficiently processed infected pork was determined as the cause of the disease. While the risk of contracting disease due to the consumption of domestic pork has been reduced or eliminated in many countries due to modern production and control procedures, for specific reasons, wild boar meat still poses a risk [1].

To date, outbreaks of trichinellosis associated with the consumption of infected wild boar meat have been reported in many European countries and the involved species were identified as *T. spiralis*, *T. britovi* [1], and *T. pseudospiralis* [2]. Infected wild boar meat was the source of outbreaks registered in France [2,3], Lithuania [4], Poland [5,6], Italy [7,8], Germany [9], Spain [10], Belgium [11], Slovakia [12], Bulgaria [13], and Romania [14].

In Croatia, according to previous epidemiological reports, the usual source of trichinellosis was pork products that caused disease in small family outbreaks [15]. In two described outbreaks [16,17] that occurred in 2002 and 2004, the authors stated that both outbreaks were caused following the consumption of smoked sausages.

In recent years in Croatia, systematic surveys and an institutionalised approach to addressing each epidemic has allowed for the examination of every single case, with the aim of determining all aspects of the registered outbreak or single cases of trichinellosis. As such, the largest outbreak in 2017 was the subject of the investigation, which included close collaboration between medical and veterinary services and was supported by the newest molecular methods in order to elucidate all the aspects of the outbreak.

This outbreak occurred in early 2017 when 26 cases with clinical symptoms were registered in two cities in two Croatian counties. The source of infections and the majority of those affected originated from the part of Croatia considered endemic for trichinellosis. Field epidemiological analyses and interviews with those affected revealed differing, contradictory, and unreliable information about the origin of the meat used to prepare the meat products. In addition, there were some suspicions that the owner of the meat products, accused of causing the outbreak, tried to conceal the identity of the offenders or hide other illegal activity. The first suspicion was related to the possible use of uninspected domestic pig meat, which is considered a serious violation of all applicable regulations, as well as the neglect of all potential ethical and health consequences of trichinellosis, which were frequent in the past, especially in this part of Croatia. Another suspicion was related to the poaching of wild boars and the avoidance of compulsory meat inspection due to illegal hunting.

In order to clarify the actual composition of the meat products, new and reliable molecular methods that can distinguish between pig and wild boar meat and/or determine the presence of other types of game meat were applied [18,19].

The aim of this study was to elucidate the circumstances of this outbreak of trichinellosis in Croatia, where the source of the disease was infected wild boar meat.

## 2. Material and Methods

Medical records: The outbreak started in early 2017, after the admission of the first patients. The clinical signs and epidemiological interview indicated acute intestinal infection and subsequent serological analysis confirmed eosinophilia. Further investigation identified additional people with similar, though milder, symptoms who had consumed homemade sausages from the same source. These findings were sufficient to raise suspicions about a potential trichinellosis outbreak. A case of trichinellosis was confirmed if a person ate suspected meat products, had pronounced clinical symptoms, eosinophilia, and/or positive serology for *Trichinella*.

IgG antibodies to the excretory secretory antigen of *T. spiralis* in the serum of patients were determined using the commercial ELISA test NovaLisa *Trichinella* spiralis IgG ELISA (NovaTec Immunodiagnostica GmbH, Dietzenbach, Germany).

Veterinary inspection: Upon receiving notice from the medical service, state veterinary inspectors visited the home of the owner of the suspected meat products, and interviewed him in detail about meat preparation and the origin of the meat. During the interviews, the owner often gave contradictory statements. The inspectors sampled and seized six meat products, prepared at the home of the owner on different dates and occasions with meat from different animal species. All meat products were sampled in the house of the owner from the facility where the products were smoke-cured and sent to the National Reference Laboratory (NRL) for further testing.

In addition, the inspectors visited the hunting ground where the wild boar samples originated from and were shot. Furthermore, the veterinary organization authorized the inspection of the meat samples in order to determine the possible omissions that led to the epidemic. In addition to the listed activities, the inspectors sealed the rest of the meat products and urgently sent them to a company for the destruction of biological waste in order to prevent the possibility of new cases of disease. All the above-mentioned activities were undertaken in accordance with the current national regulations for the suppression and eradication of trichinellosis [20].

NRL: Samples of three different types of sausages prepared on 4 December 2016, [*čajna kobasica* (salami), *kulenova seka* (pepper salami), and *kobasica* (sausage)], and of two types of sausages, [*kulen* (pepper salami) and *kobasica* (sausage)] and bacon (*slanina*) prepared on 10 December 2016, were tested using the artificial digestion method (reference method of detection) pursuant to EU Regulation No. 2015/1375 [21] and the larval burdens (LPG of sample) were determined. Three positive samples, kulenova seka from 4 December 2016, and kobasica and slanina from 10 December 2016, were detected. The larvae recovered from the three positive samples were identified by multiplex PCR [22,23].

AGES: Meat from the three positive samples was sent for further molecular analysis to confirm the absence of domestic pork and the presence of wild boar and roe deer meat, respectively, as stated by the owner in his interviews. DNA extraction was carried out using cetyltrimethylammonium bromide (CTAB) buffer in an extraction protocol [24] according to EN ISO 21571. The primers and probes for the real-time PCR experiments were already developed in previous studies [18,19]. Real-time PCR was performed in strip tubes with caps (0.1 mL, Qiagen, Hilden, Germany) on a 72-well rotor (Qiagen, Hilden, Germany) in a Rotor Gene Q cycler (Qiagen, Hilden, Germany) or on an optical 96-well reaction plate (0.2 mL, Applied Biosystems, Foster City, CA, USA) sealed with optical adhesive film (Applied Biosystems, Foster City, CA, USA) on the ABI 7500 Real-time PCR System (Applied Biosystems, Foster City, CA, USA). The PCR was conducted in a total volume of 25 µL. The reaction mix contained 12.5 µL of Quantitated Multiplex PCR NoROX Master Mix (Qiagen), 2.5 µL of ultrapure water, 5 µL of 5× primer/probe mix, and 5 µL of isolated DNA (DNA concentration between 5 and 20 ng/µL). The standard temperature program on the Rotor Gene Q cycler was initiated with a denaturation step at 95 °C for 15 min, followed by 40 cycles at 94 °C for 1 min and 60 °C for 1 min. The PCR products that resulted from the tetraplex assay for wild games species (roe deer, fallow deer, sika deer, and red deer) were detected in the dark green (emission maximum ~520 nm), light green (emission maximum ~550 nm), yellow (emission maximum ~580 nm), and red (emission maximum ~670 nm) channel, respectively.

## 3. Results

Medical facts: The outbreak started in the City of Đakovo (Osijek-Baranja County) after a young couple was admitted to a health clinic with the man experiencing stomach cramps, diarrhoea, vomiting, fever up to 38 °C, myalgia in the hands and feet, conjunctival redness, and eosinophilia, while the woman displayed somewhat milder symptoms. They recalled consuming domestic cured meat products on multiple occasions. Further investigation revealed a further 66 exposed people in Đakovo. Altogether, in Đakovo, 25 people were diagnosed with trichinellosis, 2 were hospitalised, 20 were treated as outpatients, and 4 were asymptomatic (positive serology and eosinophilia in the blood).

Several days later, one patient was hospitalised in the City of Varaždin (Varaždin County), about 220 km from Đakovo (Figure 1), with the same clinical symptoms. Anamnesis confirmed that he ate meat products brought to him by his father after hunting in Đakovo. Two more persons were determined to have been exposed in Varaždin.

A positive case is defined as a person meeting the following criteria: the consummation of suspicious meat products, pronounced clinical symptoms, eosinophilia, and/or positive serology for *Trichinella*.

In total, out of 71 exposed people, 26 were diagnosed, and 3 were hospitalised. Patients most commonly reported gastrointestinal disturbances, fever, myalgia, conjunctivitis, and eosinophilia. Of the 26 people diagnosed with trichinellosis, 17 were male and 9 were female (Figure 2).

According to the age structure, the majority (16) were between 21 and 60 years old, seven people were between 4 and 20 years old, and three were between 61 and 86 years old (Figure 3).

Hospitalised patients were treated with mebendazole 3 × 200 mg for 14 days, and others were recommended prophylaxis with mebendazole 3 × 200 mg for 5 days.

Veterinary inspection: As a result of the investigations and repeated interviews with the owner of the meat products, the owner of the hunting grounds, and authorized veterinarians, the veterinary inspectors found a shortage of 3 untested samples from the total number of 47 wild boars shot.

After sampling, the veterinary inspectors banned and prevented the use of all homemade meat products that were found on the owner’s premises until their removal and destruction in the official rendering plant.

NRL: Of the six meat products sampled and tested, the three that were identified as positive by artificial digestion were: one type of salami (*kulenova seka*) prepared on 4 December 2016 (5.5 LPG) and two meat products, bacon (*slanina*) (5.25 LPG) and sausage (kobasica) (7.08 LPG), prepared on 10 December 2016. Multiplex PCR confirmed *T. spiralis* in all three samples.

AGES: All three positive samples were tested with additional molecular methods to verify the composition of the meat in the meat products; this confirmed that all were made exclusively from wild boar meat (*Sus scrofa scrofa*) (Limit of detection, LOD = 1%). The presence of domestic pig meat (*Sus scrofa domesticus*) (LOD = 5%) and deer meat (*Cervus elaphus* and *Capreolus capreolus*) (LOD = 0.1%) was excluded.

## 4. Discussion

The seasonal slaughter and preparation of homemade meat products for personal use is still a very common tradition, especially in the rural regions of Croatia. The season usually starts in mid-November and ends by mid-February. Most households prepare meat products using pigs raised on their premises but, in recent times, some households have started to prepare meat products from pork bought from official slaughterhouses. This is a step forward in improving the level of prevention of trichinellosis in Croatia. Nevertheless, a very small number of households sometimes use smaller quantities of wild boar meat to “enhance” the flavour of meat products.

Food adulteration has become a global trend and has numerous consequences. It can pose a health risk [25] and stir up cultural and religious issues if pork meat or lard is illegally added to a product that is sold as halal [26].

However, most of the motives for food adulterations are economic [27], where the food industry tries to sell cheaper meat as a more expensive one, for any given reason. Therefore, many methods are continuously being developed to distinguish meat species and even to determine the species composition and quantification in meat [18]. One of the most challenging tasks in this field is to discriminate between wild boar and domestic pork in meat products, as the genomes of wild boar and domestic pig are highly homologous [19].

In this case, real-time PCR assays were used to help clarify two very important issues. On the one hand, we wanted to exclude the possible presence of meat from domestic pigs, as this could indicate a serious infringement of the regulations on obligatory meat inspection. In this part of the country, trichinellosis is an endemic disease.

Secondly, we wanted to narrow down the types of meat that could be causative agents of the outbreak.

According to the obtained results, we concluded that the only type of meat in the implicated meat products was wild boar meat and, therefore, this was the only possible source of the transmission.

As for why the wild boar meat was untested, we obtained an answer after several interviews with involved stakeholders in the field. In this case, the owner of the household was an employee of a hunting company. He repeatedly received the meat from hunted animals as a fee for his work or as a gift. The quantity of this meat was too large to use for cooked dishes, and, therefore, he decided to prepare meat products that could be used later in the year. The reason why, in this case, the wild boar meat escaped compulsory testing for *Trichinella*, after excluding the possible risk of poaching, was determined to be human error caused by a misunderstanding between the hunting agency and their employee, and/or by neglect. The misunderstanding was in defining who should submit the meat for testing: the organizer of the hunt or the user of the meat. In the end, both neglected the mandatory inspection of meat, which resulted in this epidemic.

Most of the affected people were from Osijek-Baranja County. The incriminated hunting agency and hunting grounds, and the place of preparing and smoking the meat products, were also in the same county. One affected man was from Varaždin County, some 200 km away (Figure 1). He became ill after eating the smoked meat products that his father had brought to him from Osijek-Baranja County after a hunting trip.

The consumption of game meat in Croatia is very low at just 0.55 kg per household member per year [28]. In the study cited, game meat included all types of game (both feathered and hairy), and it was found that the annual amount of wild boar meat consumed per household member in Croatia is extremely low.

The authors also noted a gap between the number of hunted game animals and the amount of meat consumed by the general public and supposed that only a small amount of game meat reaches the market, while hunters keep the rest. In the context of trichinellosis and wild boar meat, it is obvious that the hunters and their families and friends are most at risk for trichinellosis. Gottstein et al. [29] pointed out this hypothesis, and it was also confirmed here.

According to a recent eight-year study, the average percentage of infected wild boar with *Trichinella* spp. in Croatia was 0.17% and the majority of infected wild boars with the species *T. spiralis* were found in the eastern part of Croatia, in the counties surrounding Osijek-Baranja County [30].

However, this figure should not have any effect on human trichinellosis because all positive carcasses are to be immediately removed from the food chain by regulation. What is more, these authors found another “gap” that could strongly influence trichinellosis in Croatia; this gap is between hunted and tested wild boars, which amounted to 15,982 animals in total in the period from 2010 to 2017.

Another factor, which has the same or an even greater influence on the possibility of transmission via untested wild boar meat, is poaching. The veterinary service must report to the authorities every sample of wild boar meat submitted for examination for trichinellosis if it is not submitted with the corresponding documentation from the hunting club. This is why hunters do not submit samples of wild boars if they were shot during poaching. According to Hanley and Mikac [31], poaching is a recognised problem in Croatia, and wild boar is listed as one of the target species.

Furthermore, according to reports of the Croatian NRL [32,33], the number of registered cases of trichinellosis in Croatia in the 2010–2018 period (excluding 2017) has not exceeded 10 per year, with no patients registered in 2018. This was the first year without trichinellosis in Croatia since 1990, when official records began. According to these records, Croatia is on the way back out from the first to a second or perhaps third epidemiological pattern of trichinellosis in the European Region [34].

Even though no trichinellosis epidemics caused by wild boar meat have been previously described in Croatia, we believe that wild boar meat was and remains a potentially significant source of trichinellosis. This is due to the fact that wild boar hunting occurs in every Croatian county and that wild boar carcasses infected with *Trichinella* have been reported in most counties.

Furthermore, every year, a number of wild boar carcasses escape compulsory *Trichinella* testing [28]. On the other hand, traditional ways of consuming wild boar meat in Croatia usually include a thermal process (usually cooking but also baking), often after deep-freezing. This manner of preparation prior to consumption eliminates the risk of trichinellosis and is likely one of the reasons why there have been so few cases linked to wild boar in the past suspected to have been caused by wild boar meat in Croatia. In some suspected cases, analyses to determine the actual source of *Trichinella* larvae in the suspect meat products could not provide a definitive answer and therefore lack the scientific rigor for publication.

According to the present information, we estimate that only about 15–20% of wild boar meat is consumed after insufficient thermal processing (usually in smoked meat products). If we take into consideration that infected wild boar meat is present in many Croatian counties, and that it is possible that untested meat ends up in meat products for personal use, we can predict that new epidemics caused by infected wild boar meat can be expected in future. In particular, this scenario is expected if the mandatory inspection of meat is bypassed or if the meat is not tested using the validated reference method.

## 5. Conclusions

This is the first substantiated report of human trichinellosis in Croatia caused by the consumption of wild boar meat.The number of affected people made this one of the largest recorded trichinellosis epidemics in Croatia in the past decade.*T. spiralis* still appears to be the only etiological agent of trichinellosis in Croatia.Newly developed analytical methods (e.g., real-time PCR) are useful and reliable methods for food authentication, especially when collusion is suspected or when the composition of meat products is unknown for other reasons.

## Figures and Tables

**Figure 1 microorganisms-11-02995-f001:**
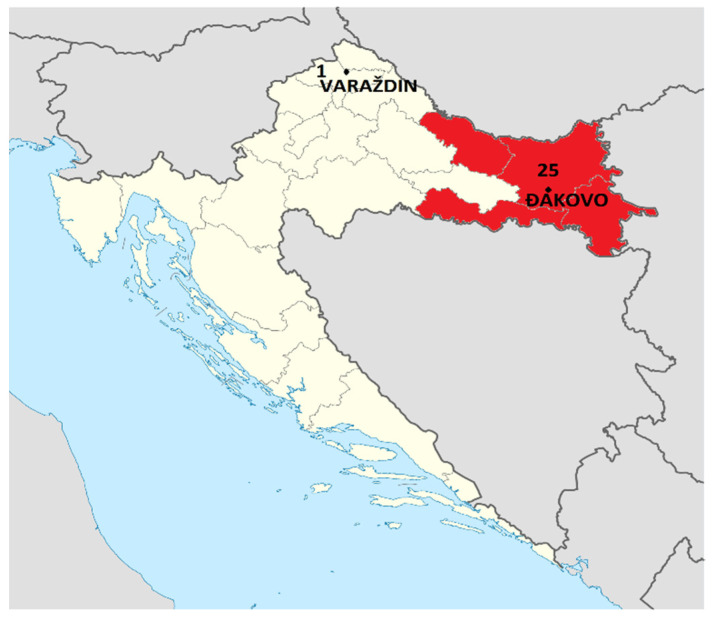
The number of affected people in two counties (in red—trichinellosis endemic counties).

**Figure 2 microorganisms-11-02995-f002:**
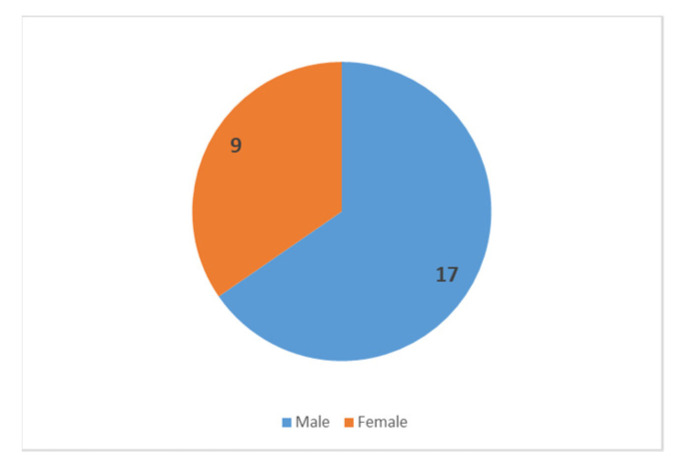
Number of registered cases by gender.

**Figure 3 microorganisms-11-02995-f003:**
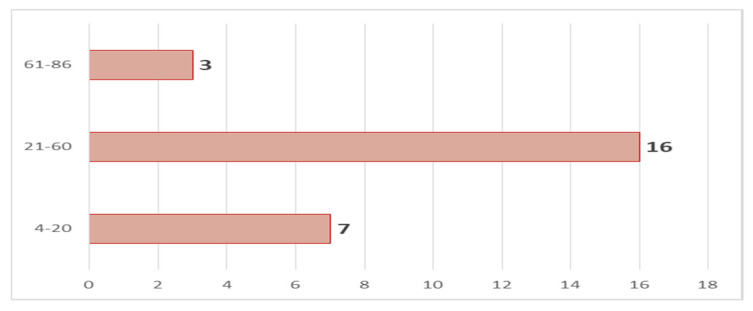
Number of registered cases by age groups.

## Data Availability

Data are contained within the article.

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
