# Peer review of "A Large-Scale Outbreak of Trichinellosis from Infected Wild Boar Meat in Croatia and the Role of Real-Time PCR Assays in Confirming the Source of the Disease"

_microorganisms, 2023, doi:10.3390/microorganisms11122995_

Round 1

Reviewer 1 Report

Comments and Suggestions for Authors

Abstract

Trichinellosis in Croatia posed a significant public health concern during the 1990s, followed by a notable improvement in the epidemiological situation. However, in 2017, there was a resurgence, with 37 recorded cases in three outbreaks and three sporadic cases. The source of this epidemic was meat products derived from wild boar, leading to 26 infections.

Paragraph Break:

Consider breaking the text into smaller paragraphs for better readability, especially when introducing new topics or methods.

Consistency in Terminology:

Maintain consistency in the terminology. For example, the text mentions "steam serology," which might be a typo. If you meant "serum serology," ensure consistency.

Clarification:

In the paragraph discussing suspicions about the meat product owner, consider providing more information or context about the suspicions and their implications. This can help the reader better understand the situation.

Please add some information about national Contingency plans according to Commission Implementing Regulation (EU) 2015/1375

Could the author provide any additional information regarding the occurrence of Trichinella spp in the surrounding countries not so far from Dakovo? This could be interesting considering the possibility of migration of wild animals, especially wild boars.

Detail Enhancement:

Consider providing more details on the "new and reliable molecular methods" mentioned in paragraph 79. This can add depth to the reader's understanding of the study methods.

Grammar and Style:

In paragraph 84, rephrase "The first patients were admitted to the hospital" for smoother readability.

Consistent Terminology:

Ensure consistent use of terminology. For example, the text alternates between "čajna kobasica (salami)" and "salami." Consistency helps in avoiding confusion.

Clarify Acronyms :

Consider briefly explaining acronyms like "NRL" (National Reference Laboratory) and "AGES" (Austrian Agency for Health and Food Safety) upon their first mention to aid comprehension.

LOD – limit of detection, LOQ?

Formatting for Readability:

Break down the text into smaller paragraphs for improved readability, especially when introducing new procedures or results.

Does the author have any additional information regarding the results of serological tests (type of test, antigens, manufacturer)? Are the laboratories using accredited methods, or are these their own developed methods?

Detail Enhancement:

Consider providing additional details about the real-time PCR process (paragraph 118) to give readers a more comprehensive understanding of the molecular analysis.

Results Section:

Introduce the results section more explicitly to guide the reader into the findings.

Clarity in Results:

Consider rephrasing the sentence in paragraph 143 for clearer communication. For example, "The final numbers were 71 exposed, 26 diagnosed, and 3 hospitalised persons" could be written more explicitly.

Consistent Terminology:

Ensure consistency in terminology. For example, "meat products" and "homemade products" are used interchangeably. Consider using a consistent term for better clarity.

Detail Enhancement:

In paragraph 152, provide more details about the actions taken by veterinary inspectors to ban and prevent the use of homemade products until their removal and destruction in the official rendering plant.

Discussion Section:

Introduce the discussion section more explicitly to guide the reader into the interpretation and implications of the results.

Consistent Terminology:

Ensure consistency in terminology. For example, the text uses both "wild boar meat" and "game meat." Consider using a consistent term throughout the text for better clarity.

Paragraph Breaks:

Consider breaking the text into smaller paragraphs for better readability, especially when introducing new topics or ideas.

Detail Enhancement:

In paragraph 208, where the reason for not testing wild boar meat is explained, consider providing more details to elaborate on the misunderstanding or neglect that occurred.

Cultural and Religious Issues:

In paragraph 180, mention is made of food adulteration stirring up cultural and religious issues. Consider providing a brief example or explanation to illustrate this point.

Discussion Continuity:

Ensure a smooth transition to the discussion section. Consider introducing the discussion more explicitly to guide the reader into interpreting the presented information.

Consistent Terminology:

Ensure consistency in the terminology used. For example, the text mentions "thermal process" and "boiling" in paragraph 255. Consider using a consistent term throughout for better clarity.

Detail Enhancement:

In paragraph 239, elaborate on the impact of poaching on the possibility of contact between untested wild boar meat and humans. Provide more information on the challenges and potential consequences.

Transition Phrases:

Add transition phrases between paragraphs, especially when moving from discussing the gap between hunted and tested wild boars to the number of registered trichinellosis cases in Croatia. This enhances the flow of the text.

Discussion Continuity:

Consider introducing the discussion section more explicitly to guide the reader into interpreting the presented information.

"meat substitution" can be considered a form of adulteration. Adulteration refers to the act of adding impurities or lower-quality substances to a product, typically to deceive consumers or reduce production costs. In the case of meat substitution, it involves replacing a certain type or quality of meat with a cheaper alternative, potentially without proper disclosure or transparency to consumers.

Practices such as poaching (unlawful hunting) and the production of processed meats from the game without paying the required fees are illegal and constitute a violation of the law. Skipping taxes fees, avoiding regulations related to hunting and game trade is unethical and can lead to many negative consequences, both for the environment and society's health.

While individuals engaged in these activities may initially profit, the long-term effects of such practices are typically harmful. Poaching can result in the depletion of wild animal populations, disrupt ecosystems, and negatively impact natural heritage or human health. Additionally, illegal activities carry the risk of legal consequences, including fines, imprisonment, and damage to reputation.

Instead, it is recommended to abide by the law, participate in legal activities related to hunting, and support sustainable management of wild animal populations. Collaboration with local environmental authorities and adherence to regulatory norms are crucial for maintaining a balance between societal benefits and nature conservation.

Literature

The literature selection is correct; however, there are more recent reports and publications on infections caused by Trichinella after consuming wild boar meat illegally hunted in Europe.

Overal nice article and interesting 

Comments on the Quality of English Language

ok

Author Response

Thank you very much for taking the time to review this manuscript. We accepted all your comments because they helped us to add more details in some parts of the article to clarify the mentioned methods or actions during the investigation. It will certainly help potential readers better understand the meaning of this article.  
In your last four comments, you emphasized the problems that can arise from food adulteration and poaching as acts of illegal activity and can have further negative consequences both for society and for individuals. I completely agree with you both in the way you explain these issues and in the consequences you found to be the results of these acts. However, I did not put them in the text, because I perceive them as your attitude and comment on the described case.

Reviewer 2 Report

Comments and Suggestions for Authors

The title must be according to the activities and results, which means write only about the determination of the composition of the meat.

 The aim description write as a separate paragraph.

Describe in more detailed the real-time PCR assays (format, primers, amplification protocol).

Conclusion number 3 could be a mistake, because by using only three samples it is not logic infer that T. spiralis is the only agent in the country.

Author Response

Thank you for taking the time to review our manuscript. Your suggestions were concise but constructive.
In the title, we wanted to highlight the role of molecular methods, which were ultimately crucial in removing all doubts related to the origin of the meat and possibly more serious criminal acts. That is why they are highlighted in the title. However, the title has also been partially corrected.
The aim of the work is separated into a separate paragraph.
Details about the PCR method - added.
Conclusion number 3 has been corrected.

Reviewer 3 Report

Comments and Suggestions for Authors

General Comments

This manuscript is generally well-written and straightforward and thus should be suitable for publication with minor revision to improve readability. Specific comments to help the authors revise the paper accordingly are suggested as follows:

Specific Comments

1. Line 2- change “massive” to “large scale” or” unprecedented”, etc.

2. Line 15- insert “high” before “public health significance” to infer that trichinellosis in Croatia continues to have public health significance.       

3. Line 19- change “cause” to “source”.

4. Lines 19-21- revise this sentence to read “Medical and epidemiological records prepared throughout the outbreak investigation and course of patient treatment were reviewed” or similar.

5. Line 21- insert “first-stage” and “(L1)” before and after “larvae”, respectively.

6. Line 22- insert “achieved” before “by artificial digestion”.     

7. Line 23- replace “by” with “using”.    

8. Line 26- total of 71 exposed persons indicated here does not agree with total of 73 accounted for in the Results section, lines 123-138 (the “couple’ in the index case and “further 68 exposed” in Dakovo, plus a hospitalized “patient” and “two more persons exposed” in Varazdin = 73). This discrepancy should be clarified/reconciled. Also change “recorded” to “documented”.

9. Line 27- change “level of infestation” to “L1 burden”.

10. Line 28- change “L/g” to “larvae per gram (LPG)”.

11. Line 30- insert “implicated” before “meat products”.

12. Line 31- replace “disease” with “occurrence”.

13. Lines 40, 41- Revise sentence as “Trichinellosis can be a serious or deadly disease in humans caused by parasites of the genus Trichinella.” This clarifies that trichinellosis is the infection/disease in humans only and does not pertain to animal infection. Thereafter in the manuscript, “trichinellosis” does not need to be preceded by “human”.

14. Line 48- change “proved” to “reported”.

15. Line 49- remove “and” before “T. britovi”.

16. Line 55- change “diseases” to “disease”.

17. Lines 56, 57- remove “principally described the medical points of outbreaks and”.   

18. Line 60- insert “in Croatia” after “years”.

19. Line 63- insert “As such” before “the largest outbreak”.

20. Lines 60-67- may want to emphasize this as a ‘One Health’ approach in this paragraph.

21. Line 68- replace “was recorded” with “occurred”. Remove “of persons”.

22. Line 76- replace “real” with “identity of the”. Remove “to” before “hide”.

23. Line 80- before “outbreak” change “the” to “this”. Remove “human”.

24. Lines 87-89- Revise sentence as “Further investigation identified additional people with similar, though milder, symptoms who had consumed homemade sausages from the same source”. 

25. Line 90- change “raising” to “raise”.

26. Line 91- replace “Finally, a case” with “A case”.

27. Line 93- remove “steam”.

28. Materials and methods- should indicate if possible how much of each sample was collected/tested by both the NRL and AGES. Also need to indicate if the methods used were performed as per the references cited with or without modifications. If modifications, need to describe them.     

29. Lines 104-112- Revise paragraph as follows: “Samples of three different types of sausages prepared on 4 December 2016 [cajna kobasica (salami)……..and kobasica (sausage)], and of two types of sausages [kulen (pepper salami) and kobasica (sausage)] and bacon (slanina) prepared on 10 December 2016, were tested using the artificial digestion method……..and the larval burdens (LPG of sample) determined. Three positive samples, kulenova seka from 4 December 2016………were detected. The larvae recovered from the three positive samples were identified by multiplex PCR…”   

30. Lines 113-118- Revise sentence as follows: “Meat from the three positive samples was sent for further molecular analysis…..in his interviews”.     

31. Line 121- change “program” to “regimen”.

32. Line 126- change “conjunctive” to “conjunctival”.

33.  Line 128- revised sentence to start as “They recalled consuming domestic cured meat products…….”.  

34. Line 138- insert “determined to have been” before “exposed”.

35. A case definition for a positive diagnosis of trichinellosis should be provided.

36. Line 142- ensure final total of 71 or 73 exposed persons is correct. Revise to read “The final numbers were xx people exposed, of which 26 were diagnosed with trichinellosis, including  three that were hospitalized.”

37. Line 145- remove “in the blood count”.

38. Line 154- replace “in the house” with “on the owner’s premises”.

39. Line 156- revise beginning of sentence as “Of the six meat products sampled, the three that were identified as positive by artificial digestion were:…..”.

40. Lines 161-164- combine first 2 sentences as “ All three positive samples tested with additional molecular methods to verify the composition……confirmed that all were made exclusively from wild boar meat……..”

41. Lines 171, 172- revise sentence to read “The season usually starts in mid-November and ends by mid-February”.

42. Line 173- replace “from their own breeding” with “raised on their premises”.

43 Line 174- remove “meat” after “pork”.

44. Line 184- replace “developing” with “being developed”.

45. Lines 184, 185- revise last part of sentence as “to determine species composition and quantification in meat”.

46. Line 195- replace “disease in humans” with “outbreak’.

47. Line 196- replace “suspected” with “implicated”.

48. Line 198- replace “the disease” with “transmission”. Insert “wild boar” before “meat”.

49. Lines 189-208- should clarify somewhere in this paragraph whether there is a  regulatory requirement for Trichinella testing of hunted wild boar, as there is for domestic pigs. It appears that there is a similar requirement for both (as per line 253 later in Discussion which indicates that wild boar testing is compulsory), but this should be clearly stated here.

50. Line 210- remove “, which is considered endemic for trichinellosis” since this is redundant, having already been stated in line 193.

51. Line 217- replace “In this study,” with “In the study cited,……”.

52. Line 220- replace “registered” with “noted”.

53. Line 222- remove “consumed” before “meat” and insert “consumed by the general public” after “meat”.

54. Line 224, 225- revise sentence as “…..hunters and their families and friends are most at risk for trichinellosis”.

55. Line 229, 230- replace “figure cannot” with “should not”, insert “to be” before “immediately”, and add “from the food chain, by regulation” after “removed”.

56. Line 232- change “the” to “this” before “gap”.

57. Lines 235, 236- change “contact between” with “transmission via”, remove “and humans”, remove “wild boar meat hunted in”.

58. Lines 242-246- not clear what is meant by his statement. Is the correct reference cited [31] after this statement?

59. Line 249- remove “human” before “trichinellosis”.

60. Line 250- replace “is present” with “occurs”.

61. Line 251- remove “the” before “wild boar”.

62. Line 254- replace “a traditional way” with “traditional ways”.

63. Line 255- replace “mostly includes” with “usually include”.

64. Line 257- insert “preparation prior to” before “consumption. Remove “this”.

65. Line 258- remove “of human trichinellosis”.

66. Line 260- insert “linked to wild boar” after “cases”

67. Line 261, 262- replace “suspicious” with “suspect”, replace “give” with “provide”, replace “detailed” with “definitive”, and replace “basis” with “rigor”.

68. Lines 264, 265- replace “an incomplete thermal process” with “insufficient thermal processing”.

69. Line 269- replace “could” with “can”, remove “the” before “future”.

70. Lines 269-272- Revise last sentence in paragraph to “In particular, this scenario is expected if mandatory inspection of meat is bypassed or if the meat is not tested using the validated reference method.”

71. Line 273- replace “proven case” with “substantiated report”.

72. Line 277- remove “is” before, and insert “appears to be” after, “still”.  

73. Line 281- replace "the people involved are not trustworthy" with "collusion is suspected".

Comments on the Quality of English Language

Quality of English is satisfactory; I have provided suggested language edits in my Comments and Suggestions for Authors above.

Author Response

Thank you for taking the time to review our manuscript.

Your suggestions were very comprehensive, precise, and useful and we accepted them all in full. All changed parts of the text are colored blue for easier tracking. We explain some of the comments in more detail below:
Comment 8: The actual number of exposed persons was 71, and a correction was made for the city of Đakovo where, in addition to the first two patients, 66 but not 68 were found to be exposed.
Commentary 28. Materials and methods chapter: the exact number of samples sampled/examined in NRL and AGES was entered. More detailed information about the methods was entered for the artificial digestion method, and for the molecular method, a request was sent to colleagues at AGES.
Comment 29: it is necessary to clarify two terms a little more: kulen and kulen seka are named for meat products that have the same composition, and the difference is in their shape, i.e. in diameter. Their appearance does not influence their potential to cause disease in humans.
Comment 35: The definition of a positive case for trichinosis was added.
Comment 49: a section on mandatory testing of wild boar meat samples for trichinella infection was added
Comment 58: Reference 31 classifies countries in Europe into 4 epidemiological categories, where the first category contains the countries with the worst epidemiological status of trichinosis, and category 4 is the best. According to the epidemiological data of the time, the improvement of the situation in Croatia and possible recategorization from the "worst" to a slightly "better" category is implied.
